# Genome-Wide Identification and Characterization of the *bHLH* Gene Family and Its Response to Abiotic Stresses in *Carthamus tinctorius*

**DOI:** 10.3390/plants12213764

**Published:** 2023-11-03

**Authors:** Zhengwei Tan, Dandan Lu, Yongliang Yu, Lei Li, Wei Dong, Lanjie Xu, Qing Yang, Xiufu Wan, Huizhen Liang

**Affiliations:** 1Institute of Chinese Herbel Medicines, Henan Academy of Agricultural Sciences, Zhengzhou 450002, China; zhwtan@126.com (Z.T.); ludandan0710@163.com (D.L.); yyl790721@126.com (Y.Y.); 15136189572@163.com (L.L.); hezixiaowei@163.com (W.D.); xulanjie18@126.com (L.X.); qingyang316@126.com (Q.Y.); 2Henan Sesame Research Center, Henan Academy of Agricultural Sciences, Zhengzhou 450002, China; 3State Key Laboratory for Quality Ensurance and Sustainable Use of Dao-di Herbs, National Resource Center for Chinese Materia Medica, China Academy of Chinese Medical Sciences, Beijng 100700, China; xiufuwan@163.com

**Keywords:** bHLH, *C. tinctorius*, abiotic stress, regulation network, ABA, salt, drought, MeJA

## Abstract

The basic helix–loop–helix (bHLH) transcription factors possess DNA-binding and dimerization domains and are involved in various biological and physiological processes, such as growth and development, the regulation of secondary metabolites, and stress response. However, the *bHLH* gene family in *C. tinctorius* has not been investigated. In this study, we performed a genome-wide identification and analysis of bHLH transcription factors in *C. tinctorius.* A total of 120 *CtbHLH* genes were identified, distributed across all 12 chromosomes, and classified into 24 subfamilies based on their phylogenetic relationships. Moreover, the 120 *CtbHLH* genes were subjected to comprehensive analyses, including protein sequence alignment, evolutionary assessment, motif prediction, and the analysis of promoter cis-acting elements. The promoter region analysis revealed that *CtbHLH* genes encompass cis-acting elements and were associated with various aspects of plant growth and development, responses to phytohormones, as well as responses to both abiotic and biotic stresses. Expression profiles, sourced from transcriptome databases, indicated distinct expression patterns among these *CtbHLH* genes, which appeared to be either tissue-specific or specific to certain cultivars. To further explore their functionality, we determined the expression levels of fifteen *CtbHLH* genes known to harbor motifs related to abiotic and hormone responses. This investigation encompassed treatments with ABA, salt, drought, and MeJA. The results demonstrated substantial variations in the expression patterns of *CtbHLH* genes in response to these abiotic and hormonal treatments. In summary, our study establishes a solid foundation for future inquiries into the roles and regulatory mechanisms of the *CtbHLH* gene family.

## 1. Introduction

Transcription factors (TFs) are pivotal in governing plant development and their adaptive responses to environmental stressors. Among these regulators, the bHLH (basic helix–loop–helix) superfamily emerges as a central player, characterized by its possession of basic and HLH domains, which are indispensable for DNA binding and dimerization [1]. The influence of the bHLH superfamily extends across a wide range of biological processes, including iron uptake [2], tanshinone biosynthesis [3], petal growth [4], stress adaptation [5], and anthocyanin biosynthesis [6]. These versatile proteins, marked by their characteristic 60-amino acid composition featuring DNA-binding basic regions and HLH hydrophobic linkages, dynamically engage in the formation of homo- or heterodimers [7,8]. Within this superfamily, various subfamilies such as E-proteins, Myc proteins, Max proteins, neurogenic, Twist family bHLH transcription factor (Twist), hypoxia-inducible factor 1-alpha (HIF-1), and single-minded homolog 1 (Sim) collectively contribute to the remarkable functional breadth and versatility of these regulators [8]. The impact of the bHLH superfamily reverberates across plant species, with 133 members identified in *Arabidopsis thaliana* (*A. thaliana*) [9], 167 in *Oryza sativa* L. *(O. sativa)* [4], and 202 in *Populus species* [10]. This ubiquity underscores their profound genetic significance. Evolutionary processes, including gene duplications, have notably enriched the repertoire of vertebrate bHLH TFs, thereby amplifying their influence on essential cellular processes [5,6].

In the intricate realm of stress response, bHLH TFs emerge as virtuoso conductors of gene expression. Their role in DNA binding, facilitated through collaborative partnerships with companion proteins, constitutes a fundamental mechanism employed by plants to counteract a myriad of challenges. They orchestrate responses to a diverse array of threats, ranging from pathogenic incursions, such as *Xanthomonas albilineans* [11], to the rigors of cold stress [12] and the formidable adversity posed by salt stress [13]. As a result, they weave a symphony of survival that extends far beyond the realm of stress management, venturing into pivotal stages of growth. They deftly choreograph embryo development [14], harmonize the formation of reproductive organs [15], and regulate both fruit and seed developmental processes [16]. Moreover, their influence extends to anthocyanin synthesis [6], light signaling [17], and brassinosteroid cascades [18]. Thus, the bHLH TFs interweave the intricate threads of both developmental intricacies and adaptive prowess, establishing connections that intertwine core regulatory pathways with stress-responsive mechanisms in plants. The purview of *bHLH* genes extends far beyond their role in stress response to encompass the orchestration of plant reactions to an array of abiotic challenges. They modulate responses to drought [7], cold [11], and iron deficiencies [14]. For example, the transcripts of the *A. thaliana bHLH122* gene exhibited significant upregulation in response to drought, high salt, and osmotic stress conditions, while there was no notable increase in response to treatment with ABA [7]. Moreover, in the leaves of *Populus species*, a fascinating response to salt stress is observed at the genetic level. In particular, certain genes, including *Potri.002G054100.1* and *Potri.002G248500.1*, display a remarkable upregulation in their expression, exceeding a two-fold increase. This substantial increase in gene expression is consistently validated via both RNA-Seq and qPCR analyses [10]. Conversely, an opposite trend is observed among other genes within the leaf, notably exemplified by *Potri.012G055700.1* and *Potri.009G117300.1*. These genes experience a notable downregulation, with their expression levels dropping by more than eight-fold. This significant reduction in gene expression highlights the complexity of the leaf’s response to salt stress, with some genes intensifying their activity while others undergo a substantial decrease in expression [10]. Nonetheless, in the study on *Passiflora edulis Sim* by Xu et al. [12], eight distinct members of the *PebHLH* gene family were examined for their expression patterns at different fruit ripening stages under various stress conditions. Among these genes, the expression of *PebHLH56* was significantly increased in response to cold stress. Researchers constructed an expression vector by combining the promoter region of *PebHLH56* with β-Galactosidase and introduced it into *Arabidopsis* plants. The experiment, as conducted by Xu et al. [12], verified the high responsiveness of *PebHLH56* to cold stress conditions in the *Arabidopsis* model.

*Carthamus tinctorius* L., commonly known as safflower or false saffron, plays a significant role within the *Compositae* or *Asteraceae* family. Flourishing mainly in arid landscapes from Southern Israel to Western Iraq, China, and India, *C. tinctorius* stands as one of the earliest annual oilseed crops, with its domestication tracing back over four millennia to the cradle of civilization, the Fertile Crescent region [19]. A testament to its enduring legacy, *C. tinctorius*’s cultivation spans vast terrains, reaching a recorded area of 717,900 hectares in 2020 and yielding an impressive harvest of 666,600 tons. The petals of *C. tinctorius*’s flowers, adorned in a captivating spectrum ranging from resplendent red to vibrant orange, bestow a precious bounty highly sought-after in the realms of both culinary and textile arts [20].

Six primary anthocyanins, namely, cyanidin, delphinidin, pelargonidin, peonidin, malvidin, and petunidin, play central roles in the determination of flower colors. Among these, the transformation of peonidin, malvidin, and petunidin is regulated through the methylation of cyanidin and delphinium pathways [21,22]. The specimens of yellow (Y) and white (W) *C. tinctorius* reveal a profusion of flavonoid metabolites, but intriguingly, the white variant lacks C-glucosylquinochalcones [23,24]. In contrast, as coloration deepens, the expression of these genes gradually diminishes [24]. The interplay of flower colors, ranging from whites (W) to yellows (Y), from light reds (LR) to deep reds (DR), has garnered considerable attention from both chemical and molecular researchers. Within the well-established framework of the core flavonoid biosynthetic symposium, a distinguished assembly of key players, including phenylalanine ammonia lyase [25], chalcone synthase [23,24], flavanone 3-hydroxylase [26], dihydroflavonol 4-reductase [27], anthocyanidin synthase [28], flavonoid glucosyltransferase [29], and anthocyanin O-methyltransferase [30]. In this study, we aimed to elucidate the role of bHLH in *C. tinctorius* and uncover their potential impact on flavonoid biosynthesis, thereby advancing our understanding of the plant’s genetic intricacies.

## 2. Results

### 2.1. Genome-Wide Identification of CtbHLH Genes in C. tinctorius

This present study identified and confirmed 120 unique CtbHLH protein sequences in the *C. tinctorius* genome, labeled from CtbHLH01 to CtbHLH120 based on chromosomal locations after filtering incomplete and redundant sequences. Notably, there was a significant variation in the distribution of these *CtbHLH* genes across the 12 chromosomes, with *CtAH09* consisting of 25 *bHLH* genes, while *CtAH12* contains only two (Appendix A). Additionally, these *CtbHLH* genes exhibited diverse characteristics, including variations in protein lengths, molecular weights, isoelectric points, instability indexes, aliphatic indexes, and average hydropathicity. Furthermore, their subcellular localization showed that 90% were located in the nucleus, 3.33% in the chloroplast, and the remaining were located in the cytosol or mitochondria (Appendix A).

### 2.2. Conserved Residues and DNA-Binding Ability Prediction of the CtbHLH Genes

This study comprehensively analyzed the phylogenetic relationships among *CtbHLH* members and revealed a diverse gene structure, with varying intron counts (ranging from 0 to 25) and coding sequences (1 to 2) among the 120 *CtbHLH* genes. Notably, *CtbHLH61* demonstrated the highest coding sequence count (27) and intron count (26), while *CtbHLH17* had no introns, featuring a single coding sequence (Appendix A). Additionally, our study identified 10 distinct motifs within the *bHLH* domains of *CtbHLH* genes, unevenly distributed among the 120 members across 19 subfamilies (Appendix A). Subfamily Ⅲe exhibited the highest motif diversity, including Motif 1, similar to *LcbHLH* domains, emphasizing its evolutionary significance [31]. Notably, Motif 9, characterized by polyG motifs, was found in several subfamilies, suggesting potential involvement in protein–protein or protein–RNA interactions. This aligns with previous research highlighting the importance of glycine-rich domains in various biological processes, including cell wall structure, stress responses, and gene regulation [32,33,34].

### 2.3. Phylogenetic Analysis and Classification of the CtbHLH Genes

To explore the evolutionary relationships within the *CtbHLH* family, a thorough analysis was performed. A phylogenetic tree was constructed based on the 120 CtbHLH proteins (Figure 1). According to the classification framework for *Arabidopsis* bHLH proteins established by Heim et al. [9] and Toled-Ortiz et al. [35], we adopted a similar strategy to classify the 120 CtbHLH protein sequences into 24 distinct subfamilies (Figure 1 and Appendix A). These subfamilies encompass a range from CtAH01 to CtAH12, following the nomenclature proposed by Heim et al. [9]. Subfamily XII had the largest number of members in safflower (17 genes), whereas Subfamily IIIa had the fewest (one gene). Although several other plant species exhibit a more extensive array of subfamilies, such as *Solanum lycopersicum* (26 subfamilies) [36] and *Brassica Rapa Ssp. Pekinensis* (26 subfamilies) [37], our investigation revealed a lower count of subfamilies in *C. tinctorius*. In particular, *C. tinctorius* presented only 24 subfamilies. Ullah et al. [38] reported the presence of 21 subfamilies in *Rosa chinensis Jacq.*, while Mao et al. [39] documented 18 subfamilies in *Malus × domestica Borkh*. Similarly, *Aquilaria sinensis* [40] and *Ginkgo biloba* [41] were found to possess 18 and 17 subfamilies, respectively. This observation suggests a relatively higher frequency of gene duplications within *C.*
*tinctorius* (Figure 1 and Appendix A). Interestingly, variations in member counts are observed across CtAH01 to CtAH12. Based on the conclusions drawn by Heim et al. [9], the coexistence of CtbHLH proteins within a shared subfamily suggests potential functional resemblances.

### 2.4. Chromosomal Location and Collinearity Analysis of CtbHLH Genes

Gene duplications are crucial for the expansion of protein-coding gene families in plants, encompassing various events, such as whole-genome, dispersed, tandem, proximal, and singleton duplications [42]. In the case of the *C. tinctorius* genome, our analysis of collinear blocks revealed 41 gene pairs, with 24 of them located on different chromosomes, indicating segmental duplications might contribute to the expansion of the *CtbHLH* gene family (Figure 2). Interestingly, only one tandem duplication gene pair, *CtbHLH99* and *CtbHLH100* on *CtAH10*, was identified (Figure 2). This pattern of duplication events aligns with the limited number of subfamilies observed in *C. tinctorius*, demonstrating similar evolutionary trends seen in *Populus deltoids* [43] and *Gossypium hirsutum* [44].

To determine the evolutionary relationships among bHLH TFs across different species, collinearity plots were generated to indicate the correlation between *C. tinctorius* and *A. thaliana*, *C. tinctorius*, and *O. sativa*, as well as *C. tinctorius* and *Helianthus annuus* (*H. annuus*) (Figure 3). Through this collinear analysis, we found that there were 102 orthologs shared between *C. tinctorius* and *A. thaliana*, 15 orthologs between *C. tinctorius* and *O. sativa*, and 200 orthologs between *C. tinctorius* and *H. annuus* (Figure 3). These findings underscore a considerable evolutionary divergence and an expansion of the gene family preceding the branching of these three species. Furthermore, the prevalence of numerical correspondences within the collinear relationships highlights the conserved nature of *CtbHLHs*. These findings imply that the collinear *CtbHLHs*, distributed among various species, may share a common ancestral origin (Figure 3).

### 2.5. Analysis of Cis-Acting Elements of CtbHLH Gene Family

In the *CtbHLH* gene promoter regions, we identified and systematically classified cis-regulatory elements into three main functional domains: plant growth and development, phytohormone responsiveness, and abiotic and biotic stresses (Appendix A). Notably, specific motifs, such as the drought response element (ABRE) (*n* = 631) and the salicylic acid-responsive TCA element (*n* = 114), were highlighted, with significant roles in ABA-dependent signaling [45] and the salicylic acid pathway [46]. Furthermore, we observed the prevalence of motifs like the G-box (*n* = 564), associated with responses to various signals, and light-responsive elements such as Box4, GT1-motif, TCT-motif, and MSL recognition elements, which play essential roles in plant growth and development [47]. Additionally, for stress response, elements such as the anoxic response element (ARE) (*n* = 321) significantly contribute to MeJA-responsiveness [48], and the CGTCA-motif (*n* = 213) holds importance as a vital component of the regulatory network, including cAMP-regulated enhancement [49].

### 2.6. Protein Interaction Network of CtbHLH Genes

The protein interaction network of CtbHLHs is a key focus, where 19.17% of paired nodes exhibit varying levels of coexpression, with the highest coexpression observed between CtbHLH100 and CtbHLH99, and the lowest between CtbHLH32 and CtbHLH69. Importantly, no co-expression and gene fusion events were detected, and there was a lack of interactions among chromosomes, phylogenetic co-occurrence, and database annotations, suggesting no associations with translation or co-translational degradation. Homologous interactions accounted for 12.50% of paired nodes, and interactions constituted 15.83% of the paired nodes. Key nodes within this network, such as CtbHLH32, CtbHLH72, CtbHLH85, and others, may regulate mechanisms related to plant growth, phytohormone responsiveness, and stress responses potentially via the formation of homodimers or heterodimers. Correlation analysis confirmed significant interactions among all 120 CtbHLH (Figure 4).

### 2.7. Expression Profiles of CtbHLH Genes

Significant variations in the expression patterns of *CtbHLH* genes were observed across diverse flower colors (Figure 5A), organs, and stages of plant development (Figure 5B). In terms of flower color, the expression values of white, yellow, light red, and dark red flowers were 10.10 ± 19.70, 10.57 ± 18.63, 11.35 ± 21.38, and 11.04 ± 24.90, respectively (Figure 5A, Appendix A). These analyses identified specific genes with the highest mean expression levels for each color: *CtbHLH42* (111.02) for white, *CtbHLH88* for yellow, light red, and dark red (92.39, 107.15, and 172.82, respectively) (Appendix A). Notably, 20.00% of white flower samples, 24.17% of yellow flower samples, 21.67% of light red flower samples, and 20.00% of dark red flower samples exhibited mean expression levels exceeding 10 (Figure 5A, Appendix A).

Furthermore, the genetic mechanisms governing various aspects of *C. tinctorius* development, encompassing seed formation, germination, and flower development, were elucidated. Within seed formation, an intricate gene expression pattern emerged, with 29 genes initially upregulated in early ovaries, dwindling to 10 during seed development at DAF10 but resurging to 17 by DAF20. Notably, 13 genes exhibited downregulation from DAF0 to DAF20, while 16 genes displayed a dynamic pattern of upregulation and downregulation, with *CtbHLH85* peaking at DAF10 and *CtbHLH23* and *CtbHLH94* showing high expression levels at DAF20. Seed germination analysis uncovered a dynamic upregulation pattern, with 40 genes upregulated at 1DAG, 24 at 3DAG, 26 at 5DAG, and 21 at 7DAG. Conversely, 12 genes remained downregulated, emphasizing their roles in early cotyledon-related processes. Additionally, 7 genes displayed upregulation at later germination stages, and 21 genes exhibited a fluctuating expression pattern. The flower development phase involved distinct stages, with genes upregulated in each and a set of 12 genes consistently downregulated. Further exploration into the genetic regulation of high linoleic acid (LA) content involved two cultivars, ‘HL’ and ‘LL,’ at different time points. The analysis revealed unique gene expression patterns associated with LA content. These findings provide valuable insights into the intricate molecular mechanisms governing *C. tinctorius* development, flower color, and oil content regulation, underlining the complexity of these processes (Figure 5B, Appendix A).

### 2.8. Expression Analysis of CtbHLH Genes under Abiotic Stress

The responses of *C. tinctorius* leaves to various stressors were also assessed, revealing distinctive patterns of gene expression in response to salt stress (NaCl), drought stress (PEG6000), and hormone treatments (ABA and MeJA). Notably, 15 genes demonstrated significant changes in expression profiles over a 24 h period in response to these stress treatments. In the case of ABA treatment, genes such as *CtbHLH*12 and *CtbHLH*26 exhibited substantial upregulation, with *CtbHLH*12 reaching a peak expression level of 8 at 24 h. Interestingly, *CtbHLH*99 and *CtbHLH*119 displayed heterogeneous expression patterns with distinct peak timings (Figure 6A). When exposed to salt stress (NaCl), a group of genes, including *CtbHLH*12 and *CtbHLH*26, showed notable upregulation, with *CtbHLH*12 reaching a peak level of 20. In contrast, *CtbHLH*74 exhibited consistent downregulation, while *CtbHLH*88 and *CtbHLH*105 displayed bell-shaped expression patterns, peaking at 6 h (Figure 6B). Under drought stress (PEG6000), genes such as *CtbHLH*12, *CtbHLH*20, and *CtbHLH*26 showed significant upregulation, with *CtbHLH*26 reaching a peak expression level of 50. In contrast, *CtbHLH*24 exhibited downregulation, and *CtbHLH*70 and *CtbHLH*99 displayed bell-shaped expression patterns with distinct peak timings (Figure 6C). Finally, exposure to MeJA resulted in more pronounced gene expression changes. *CtbHLH*6 and *CtbHLH*105 exhibited significant upregulation with peaks at 24 h. *CtbHLH*12 and *CtbHLH*119 displayed bell-shaped expression patterns with peaks at 12 h (Figure 6D). These findings underscore the dynamic and stress-specific responses of *CtbHLH* genes in *C. tinctorius* leaves, shedding light on the complex regulatory mechanisms involved in stress adaptation (Figure 6).

## 3. Discussion

In this present study, we conducted a whole genome analysis of 120 *CtbHLH* genes in *C. tinctorius*. These genes were systematically categorized into 12 distinct subfamilies based on the presence of specific conserved amino acids and other structurally conserved domains (Appendix A). Interestingly, when comparing the number of *CtbHLH* genes in *C. tinctorius* to other plant species, we found that the number of *CtbHLH* genes in *C. tinctorius* was relatively similar to that of *Capsicum annuum* L., which encompasses 122 genes classified into 21 subfamilies [50,51]. However, the number of *bHLH* genes in *C. tinctorius* is smaller compared to other plant species [52]. For instance, *Solanum tuberosum* L. has 124 *bHLH* genes organized into 15 subfamilies [53]. In *Cucumis sativus* L., a total of 142 *bHLH* genes were classified into 32 subfamilies, indicating a higher level of gene diversity [54]. Similarly, *Phaseolus vulgaris* L. harbors 155 *bHLH* genes distributed across 21 subfamilies [55]. The *Oryza sativa* L. genome has 167 *bHLH* genes divided into 22 subfamilies [4]. Similarly, *Malus × domestica* contains 188 *bHLH* genes that are classified into 18 subfamilies [39]. A larger number of 218 *bHLH* genes in 20 subfamilies was identified in *Chenopodium quinoa* (*C. quinoa*) [52]. Conversely, the *C. tinctorius* genome exhibits a higher count of *bHLH* genes compared to *Vitis vinifera* L., with 94 genes distributed across 15 subfamilies [53], *Fragaria vesca*, with 113 genes grouped into 26 subfamilies [10], and *Ziziphus jujuba Mill.*, with 92 genes sorted into 16 subfamilies [56]. This comparison highlights the variations in *bHLH* gene numbers and distributions across diverse plant species, underscoring the dynamic nature of gene families in the plant kingdom.

The existence of introns and exons can also contribute to the functional diversification of gene families. In our study, some *CtbHLH* members, such as *CtbHLH17* in Subfamily IVa (Appendix A), exhibited few or no introns, which might be associated with higher expression levels in plants [57,58]. Interestingly, the members of Subfamily IV showed the least number of introns, a pattern similar to Group D of *Aquilaria sinensis* [40]. This suggests that *CtbHLHs* in these subgroups may enable rapid and efficient responses to various stresses [59]. Although the intron distribution patterns observed in our study are dissimilar to those of other plant species, such as *Capsicum annuum* L., *Panax ginseng*, and *Malus* × *domestica* [51,58,60], these differences underscore the complex and diverse evolutionary paths of various plant species, resulting in specific genomic characteristics and gene regulatory mechanisms. In our investigation of *bHLH* gene members across various plants, we conducted a thorough comparison of gene structures. The results highlighted notable differences in the distribution of CtbHLH members among different subfamilies. Interestingly, our findings revealed that Subfamilies IIII and VI lack any CtbHLH members. In contrast, Subfamily XII exhibited the highest count of CtbHLH members, containing 17 members (Appendix A). This distribution pattern of *bHLH* gene family in *C. tinctorius* aligns with observations from other plants, such as *Citrus grandis* (*C. grandis*), where Group 1 was the most extensive subfamily with 17 CgbHLH members, while Group 3 comprised only 2 members [14].

The presence of distinct bHLH members in these subfamilies often corresponds to their involvement in specific biological roles. Subfamily I members in plants are known to exhibit diverse functions, including cold adaptation [61], protection against cell differentiation [62], the regulation of flower development [63], as well as response to cytokinin [64] and jasmonic acid [65] stimuli, and tapetal layer and anther development [48]. On the other hand, members of Subfamily X, found in both *C. tinctorius* and *C. grandis*, have been associated with stomatal complex development [66].

Although the numbers of detected bHLH members in *C. tinctorius* and *C. grandis* were relatively high, surpassing the count found in *Ginkgo biloba* (*G. biloba*) [41], a noteworthy similarity between *C. tinctorius* and *G. biloba* is their absence of members in Subfamily VI [41]. This shared characteristic suggests a potential evolutionary loss in these proteins during the development of *C. tinctorius*. Our systematic analysis of bHLH gene distribution across various subfamilies within the *C. tinctorius* genome offers valuable insights into their functional diversity. Comparisons to other plant species further enhance our understanding of the evolutionary patterns that have influenced the composition and functions of the bHLH gene family. The CtbHLH family members, except for the Subfamilies I and X, also play distinct functional roles in various flowering plants. For example, some *bHLH* genes have been implicated in the abiotic stress resistance and reproductive development of *Chenopodium quinoa*. Notably, the expression levels of *CqbHLH88* and *CqbHLH144* have been found to potentially impact abiotic stress tolerance in *C. quinoa* [52]. These genes reach peak expression on the 21st day after flowering, highlighting their involvement in these critical developmental processes [52]. Similarly, in other plants, such as *Cucumis sativus*, specific bHLH genes have been identified as pivotal regulators in response to abiotic stresses. For instance, the overexpression of *CsbHLH041* governed by the 35S promoter enhanced the tolerance of transgenic Arabidopsis plants and *Cucumis sativus* seedlings to both salt and ABA stresses [54].

The variation in the number of *bHLH* genes across different plant species can be attributed to gene duplication events, genome size, and gene loss during the course of evolution [54]. In this study, we examined the conserved motifs of *CtbHLH* genes to reveal their genetic and functional characteristics. This analysis reveals that CtbHLHs within the same subfamily share similar genetic and motif structures, confirming the accuracy of the subgroup classification in the phylogenetic tree (Appendix A). Among the identified motifs, motif 1 (ERRRLLP) was detected in nearly all CtbHLH proteins. These motifs are integral components of the bHLH domain, known for their high conservation and significant implications for DNA binding [9,35]. However, the eight conserved non-bHLH domains, except for the bHLH domain, also appeared in CtbHLHs across their respective subfamilies. This observation aligns with the findings from other plant species, such as *Capsicum annuum* L. [51], *Chenopodium quinoa* [52], and *Cucumis sativus* [54]. In our study, a distinct subfamily exhibited motifs 8, 9, and/or 10. These motifs are associated with bHLH-MYC and R2R3-MYB TFs N-terminal domains. Members within this subfamily may act similarly to those found in *Panax ginseng*, where these factors are involved in regulating phenylpropane biosynthesis [60]. The interaction between MYB and bHLH TFs has been shown to influence various processes, including defense metabolism, anthocyanin biosynthesis, and organ development, such as trichome initiation [67,68,69,70].

Subfamilies Ia and Ib of *bHLH* genes, where *CtbHLH26* and *CtbHLH74* showed a common response to ABA and salt stress, as indicated in the phylogenetic tree and expression profiles (Figure 2 and Figure 5A,B). This finding is specific to *C. tinctorius*, and similar stress responses might involve different subfamilies in other plant species. For instance, *A. thalia* and *Beta vulgaris* utilize *AtbHLH92* and *BvbHLH92*, respectively, to enhance tolerance to salt and osmotic stresses, partially dependent on ABA signaling [71]. Additionally, the overexpression of *OsbHLH068* in *A. thalia* has been linked to reduced salt-induced hydrogen peroxide accumulation [72]. Similarly, the expression of *TabHLH13* in *Triticum aestivum* L. is increased with increasing salt ion concentrations [73]. Both *CtbHLH6* (Subfamily IIIe) and *CtbHLH105* (Subfamily IIIf) were upregulated in *C. tinctorius* following MeJA treatment, indicating their potential roles in MeJA-related responses (Figure 2 and Figure 5A,B). However, in *Dendrobium huoshanense*, members of Group IIIe, *DhbHLH81* and *DhbHLH20* exhibited high expression levels at 16 and 4 h of MeJA treatment, respectively, surpassing the baseline levels by more than 20 times [74]. The results for Subfamilies Ia, Ib, IIIe, and IIIf highlight how different species employ distinct subfamilies to respond to specific stress treatments.

The prevalence of cis-regulatory elements, particularly *ABRE* elements and G-box motifs, within the *CtbHLH* genes may affect their responsiveness to stress conditions. As illustrated in Figure 6A, the *CtbHLH88* gene from Subfamily XII is upregulated following ABA treatment at the 24 h interval and reaches a peak expression at 6 h after NaCl treatment (Figure 6B). It is worth mentioning that *CtbHLH88* is enriched with 28 ABRE elements and 27 G-box motifs (Appendix A). *CtbHLH99*, which contains 28 ABRE elements and 30 G-box motifs (Appendix A), presents a heterogeneous pattern of both upregulation and downregulation during the 24 h period after ABA and drought treatments (Figure 6A,C) and is relatively susceptible to salt stress (Figure 6B). Intriguingly, the presence of ABRE-ABRE pairs in both genomes indicates their potential to form functional ABA-responsive complexes in *A. thalia* and *Oryza sativa*, thereby facilitating stress-related gene regulation [75]. Additionally, previous studies have shown the significance of G-box elements in stress response mechanisms. For instance, *ZmPTF1* has been identified as a regulator of drought tolerance in maize, influencing root development and ABA synthesis [76]. *ZmPTF1* binds to the G-box elements in the promoters of various genes, such as (*9-cis-epoxycarotenoid dioxygenase*) *NCEDs*, *C-repeat binding factors* (*CBF4*), *NAC081*, and NAC domain protein (NAC30), thereby activating their expression and contributing to drought adaptation [76].

A subset of genes, encompassing *CtbHLH88*, *CtbHLH99*, *CtbHLH12*, *CtbHLH26*, *CtbHLH102*, *CtbHLH24*, *CtbHLH105*, and *CtbHLH6*, are characterized by the presence of MBS elements (*n* = 1 each) (Appendix A). Compared to the findings in other plants [28,77], these *bHLH* members demonstrate an array of defense and stress-responsive elements, including the drought-inducible MBS element. In comparison to G-box and ABRE, MBS elements might hold a lesser significance than G-box elements. Notably, in *C. tinctorius*, *CtbHLH88* exhibits a diverse expression pattern throughout different developmental stages, spanning from DAF to DES and of varying flower colors, with a peak during PFS (Figure 5B). This gene also shows distinct upregulation patterns across four different flower colors (Figure 5A). Similarly, *CtbHLH6* is upregulated during DAF (Figure 5B) and exhibits distinct upregulation patterns across the four flower colors (Figure 5A). Both *CtbHLH12* and *CtbHLH26* are upregulated in response to salt and drought treatments (Figure 6B,C), with *CtbHLH26* upregulated following ABA treatment (Figure 6A). The G-box binding factor specifically binds to G-rich elements within early post-aggregative genes, which can be activated via cAMP [78].

*CtbHLH102* and *CtbHLH12* are distinguished by their upregulation patterns in response to drought stress and their unique bell-shaped gene expression patterns following MEJA treatment, respectively (Figure 6C,D), and may be important candidates due to their abundance of TGACG_elements (*n* = 5) across all members. Although *CtbHLH102* is downregulated from 1DAG to 10DAG and from SBS to DES (Figure 5B), its significant 25 G-box elements and 22 ABRE elements (Appendix A) contribute to its distinct upregulation pattern under drought stress (Figure 6C). From the analysis of the *CtbHLH74* gene (Appendix A), the abundance of TGACG elements (*n* = 5) is higher compared to ABRE (*n* = 3), and MBS (*n* = 2) elements are aligned with a 50% downregulation at the 24 h time interval following salt treatment (Figure 6B). The TGACG elements, which have been identified within the *PR* gene sequences of *A. thaliana* and *O. sativa*, are responsive to methyl jasmonate and modulate the transcription of PR sequences by binding with BZIP TGA factors [79]. Furthermore, the presence of TC-rich repeats in genes such as *AtPR1*, *AtPR2*, *OsPR2*, and *OsPR9* underscores their roles in stress and defense responsiveness.

## 4. Materials and Methods

### 4.1. Identification of the bHLH Genes from C. tinctorius

The genome sequences of *C. tinctorius* were obtained from the Genome Database [80,81]. To comprehensively identify the CtbHLHs, the hidden Markov model (HMM) profile of the bHLH domain (PF00010) from the Pfam database [82,83] was used to search for bHLH protein members in the *C. tinctorius* protein sequence file using HMMER 3.0 with default parameters (E-value cut-off < 10^−5^). Moreover, 158 AtbHLH protein sequences were obtained from The Arabidopsis Information Resource (TAIR) [84]. Blastp v2.12.0 was used to identify potential *bHLH* genes in the *C. tinctorius* genome (E-value < 10^−5^). After combining and removing redundancies, the conserved domains of all CtbHLH proteins were determined using Batch CD-search [85,86], Pfam [82], and SMART [87,88]. Additionally, the ProtParam tool from the ExPASy website [89,90] was employed to predict the sequence length, molecular weight, and isoelectric point of the identified CtbHLH proteins. Finally, the subcellular localization of CtbHLH proteins was predicted using WoLF PSORT [91].

### 4.2. Chromosomal Locations, Multiple Alignment Analysis and Phylogenetic Analysis

The *CtbHLH* genes were mapped to 12 chromosomes using the *C. tinctorius* genome annotation GFF3 file, which contains positional and gene structure information. The mapping procedure was executed with MapGene2Chrom [92,93]. To align the CtbHLH and AtbHLH proteins, a multiple sequence alignment was performed using ClustalW 2.0 according to a previously described method [94]. TBtools v1.130 software [95] was employed for the visualization and analysis of the conserved domains in CtbHLH proteins. Then, a phylogenetic tree was constructed using MEGA 7.0 with the neighbor-joining method, utilizing parameters such as 1000 bootstrap replicates, the Poisson model, and pairwise deletion. Based on the classification of closely related AtbHLHs and the bootstrap support values at corresponding nodes, all CabHLHs were divided into subfamilies.

### 4.3. Gene Structures, Conserved Motifs and Promoter Analysis

The exon/intron structures of each *CtbHLH* gene were analyzed using TBtools v1.130 [95]. To identify conserved motifs, the MEME-Suite 5.1.1 online program [96,97] was employed with specific parameters: the recognition motif limit was set to 10, the minimum motif width was set to 6, and the maximum motif width was set to 50. To examine regulatory elements, the 2000-bp sequence upstream of the ATG start codon was extracted using TBtools v1.130 [95], and all promoter sequences of *CtbHLH* genes were submitted to the PlantCARE database [98] for cis-acting element prediction.

### 4.4. Gene Duplication and Collinearity Analysis

Gene duplication events were analyzed using Blastp v2.12.0 with default parameters, while the detection of collinearity relationships between *C. tinctorius* and two other species (*O. sativa* and *A. thaliana*) was performed using TBtools v1.130 [95].

### 4.5. Protein Interaction Network Analysis

The interaction among CtbHLH proteins, referencing to AtbHLH protein, was analyzed using STRING [99], with the parameter threshold set to 0.15. The resulting network was visualized using Cytoscape v3.8.2 [100].

### 4.6. Expression Analysis of CtbHLH Genes

RNA-Seq data from the National Center for Biotechnology Information (NCBI) BioProject database (accession number PRJNA646045) were used to investigate the expression of *CtbHLH* genes in seedlings, seeds, and flowers at various developmental stages [81]. In particular, cotyledons were sampled at 1, 3, 5, 7, and 10 days after germination (DAG), while filament samples were taken at five different stages: small bud stage (SBS), middle bud stage (MBS), initial flowering stage (IFS), peak flowering stage (PFS), and decayed flowering stage (DFS) during flower development. Seeds from two cultivars were collected: one cultivar (‘HL’) with high linoleic acid (LA) content and the other cultivar (‘LL’) with low LA and high oleic acid (OA) content. The seed samples were collected 10 and 20 days after flowering (DAF). Furthermore, RNA-Seq data from the NCBI BioProject database (accession number PRJNA738310) were utilized to determine the expression of *CtbHLH* genes in four *C. tinctorius* materials with distinct colors, namely, W, Y, LR, and DR.

### 4.7. Plant Materials and Treatments

*C. tinctorius* (cv. Yuhonghua No. 1) plants with full and uniform-sized seeds were selected and sowed in pots. After 6 days of germination, the seedlings were transferred into Hoagland nutrient solution for hydroponic culture. Subsequently, the plants were grown in a growth chamber for two weeks at 25 °C under a photoperiod of 16:8 h light/dark cycle.

To investigate the expression levels of candidate *CtbHLH* genes under different stresses and hormone treatments, 2-week-old *C. tinctorius* seedlings were transferred to Hoagland nutrient solution containing 200 mM NaCl (salt stress), 10% PEG6000 (drought stress), 100 μM abscisic acid (ABA) or methyl jasmonate (MeJA). Following these treatments, the leaves were sampled at time points of 0, 3, 6, 12, and 24 h. Three biological replicates were taken for each treatment.

### 4.8. Expression Analysis of the CtbHLH Genes by qPCR

Total RNA was isolated using the Quick RNA Isolation Kit (HuaYueYang, Beijing, China) following the manufacturer’s instructions. The PrimeScript^TM^ RT reagent Kit with gDNA Eraser (TaKaRa, Beijing, China) was used for first-strand cDNA synthesis according to the manufacturer’s instructions. Quantitative real-time PCR (qPCR) was performed using the SYBR^Ⓡ^ Green qPCR Mix (Monad, Suzhou, China) on the QIAquant 96 2plex real-time Detection System (QIAGEN, Hilden, Germany). The qPCR cycling conditions were as follows: initial denaturation at 95 °C for 30 s, followed by 40 cycles of denaturation at 95 °C for 10 s and annealing/extension at 60 °C for 30 s. The *Ct60S* gene was used as a housekeeping gene, and all reactions were conducted in three biological replicates. In their *C. tinctorius* qPCR experiment, Tu et al. [101] employed a housekeeping gene, a choice later validated by Liu et al. [102]. For the analysis of expression levels in response to abiotic stress conditions for the target *CtbHLH* genes, normalization was conducted against the chosen housekeeping gene. Relative expression levels were then determined using the 2^−ΔΔCt^ method. The primer sequences used in this study are listed in Appendix A.

### 4.9. Statistical Analysis

Statistical significance was determined using Student’s *t*-test, with “*” indicating *p* < 0.05 and “**” indicating *p* < 0.01. The expression levels at each time point were compared between the control and NaCl, PEG6000, ABA, or MeJA treatment groups.

## 5. Conclusions

In this study, we identified 120 *CtbHLH* genes distributed among 24 subfamilies in *C. tinctorius*, unraveling their diverse roles, especially in stress responses. Subfamilies VII had the most CtbHLH members, while Subfamilies Ib, IVb, VIIa, and X had the fewest. Notably, *CtbHLH6* and *CtbHLH105* from the corresponding Subfamilies IIIe and IIIf were upregulated following MeJA treatment, suggesting their involvement in MeJA-related responses. We found that ABRE elements and G-box motifs played significant roles in regulating stress responses. *CtbHLH88*, enriched with 28 ABRE elements and 27 G-box motifs, was upregulated after ABA and NaCl treatments. Similarly, *CtbHLH99*, with 28 ABRE elements and 30 G-box motifs, responded positively to salt stress. *CtbHLH12* and *CtbHLH26* were upregulated in response to salt and drought treatments, while *CtbHLH26* responded to ABA treatment only. In addition, we identified *CtbHLH102* and *CtbHLH12* as key players due to their abundance of TGACG_elements. *CtbHLH102* was upregulated during drought despite initial downregulation, attributed to its 25 G-box elements and 22 ABRE elements. *CtbHLH74*, with more TGACG elements, was significantly downregulated after salt treatment. *CtbHLH88* displayed diverse expression patterns across developmental stages and flower colors, peaking during the flowering stage and upregulating across various flower colors. This study reveals the intricate nature of stress response networks in *C. tinctorius*, enhancing our understanding of plant stress responses and potentially contributing to the development of stress-tolerant crops.

## Figures and Tables

**Figure 1 plants-12-03764-f001:**
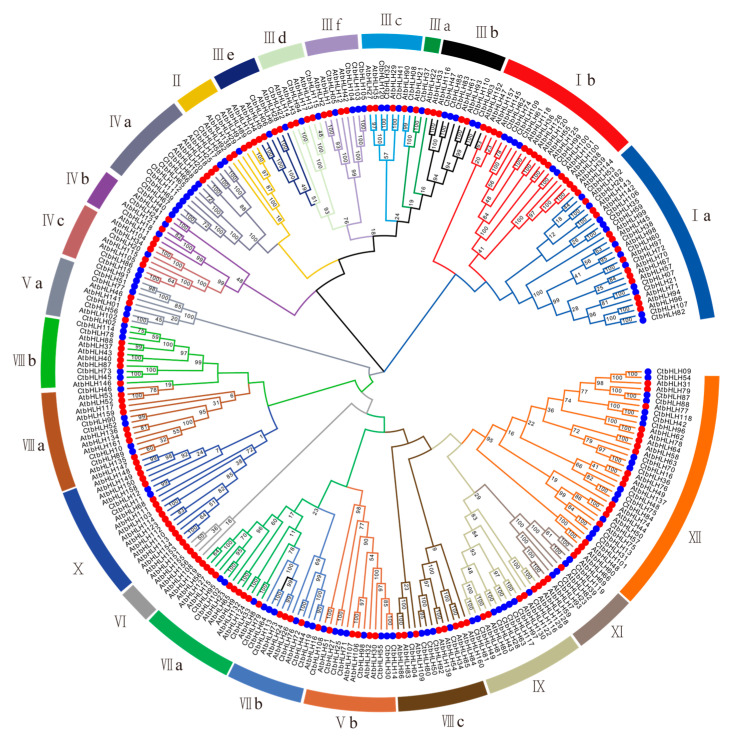
Phylogenetic tree and classification of bHLH subfamily proteins in *A. thaliana* and *C. tinctorius* were generated using MEGA 7.0 with 1000 bootstrap replicates. Different colored branches indicate different subgroups. Red circles represent AtbHLH proteins, and blue circles represent CtHLH proteins. Roman numerals line up with the bHLH subfamily.

**Figure 2 plants-12-03764-f002:**
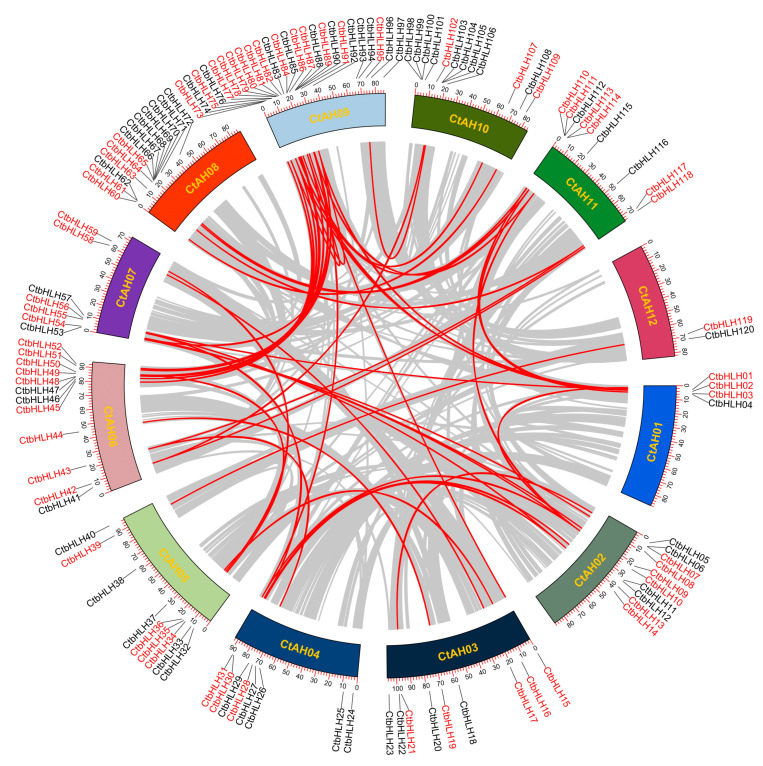
Distribution of 120 *CtbHLH* genes on 12 *C. tinctorius* chromosomes, and the syntenic map among *bHLH* gene family from *C. tinctorius*, analyzed using TBtools v1.130. Gray lines in the background indicate the synteny blocks within the *C. tinctorius* genome. The syntenic *CtbHLH* gene pairs are marked with red lines and highlighted in red.

**Figure 3 plants-12-03764-f003:**
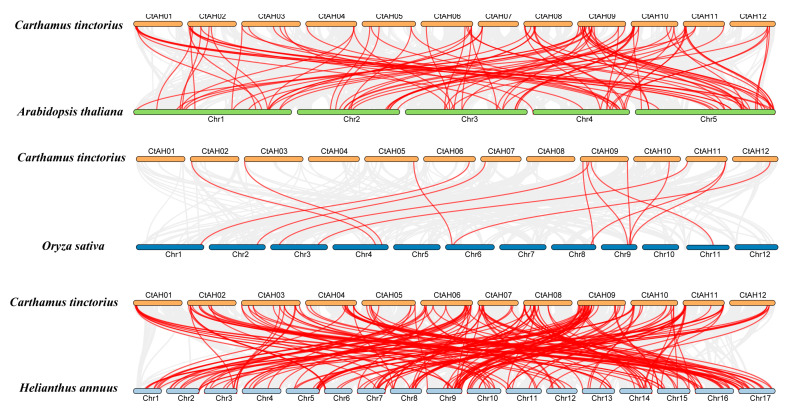
Collinearity relationships of *CtbHLH* genes between *C. tinctorius* and *A. thaliana*, *C. tinctorius* and *O. sativa*, and *C. tinctorius* and *H. annuus*, constructed using TBtools v1.130. The identified orthologous *bHLH* genes are connected by red lines. Gray lines in the background indicate the colin-ear blocks between two plant genomes.

**Figure 4 plants-12-03764-f004:**
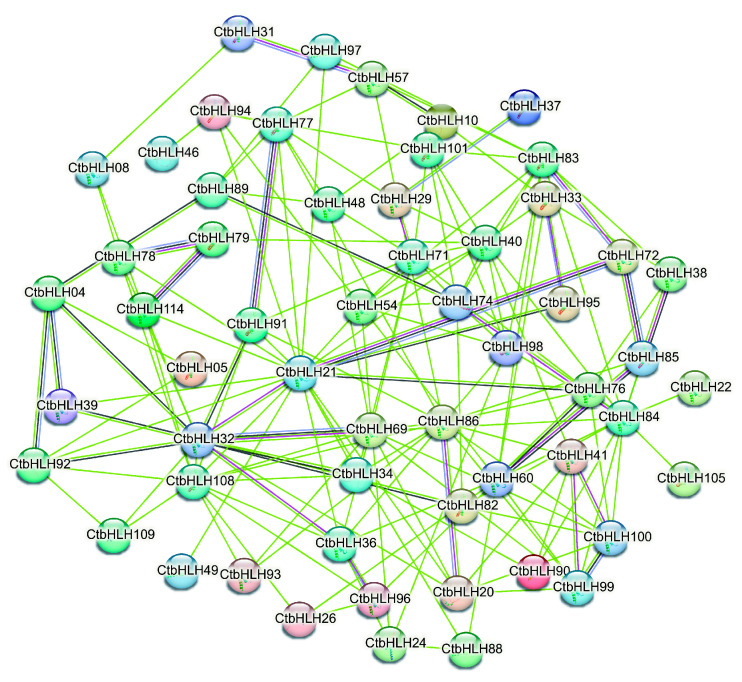
Protein interaction of CtbHLHs based on the STRING analysis.

**Figure 5 plants-12-03764-f005:**
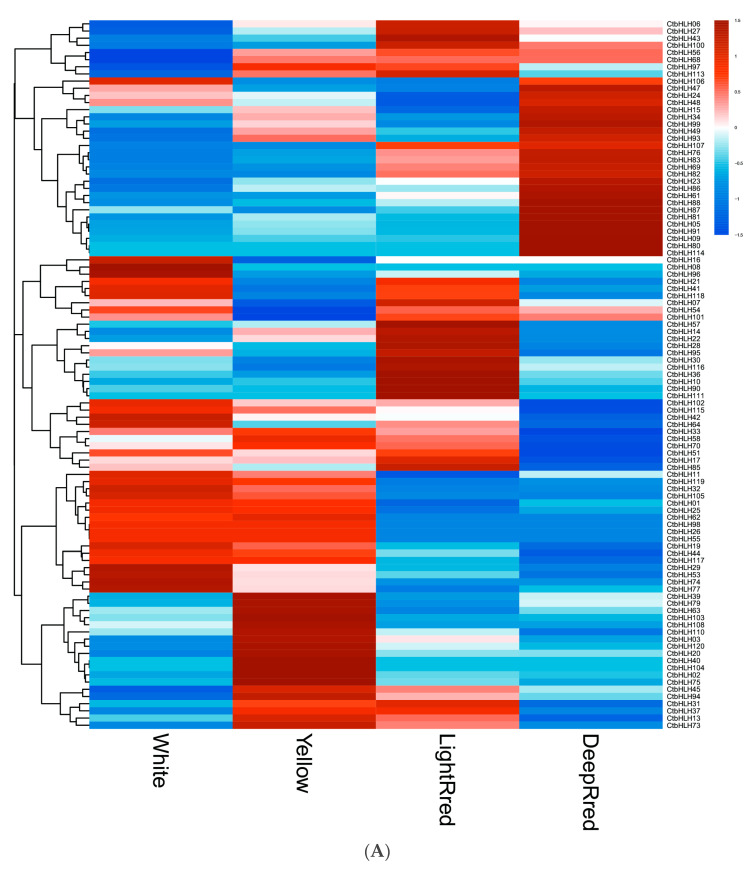
RNA-seq data from NCBI. Heatmap of *CtbHLH* gene expression in *C. tinctorius*. (**A**) Gene expression analysis of *CtbHLHs* in relation to four different flower colors in *C. tinctorius.* (**B**) Gene expression analysis of *CtbHLHs* in relation to different stages in *C. tinctorius* development. Dark orange color represents upregulation, and dark blue color represents downregulation.

**Figure 6 plants-12-03764-f006:**
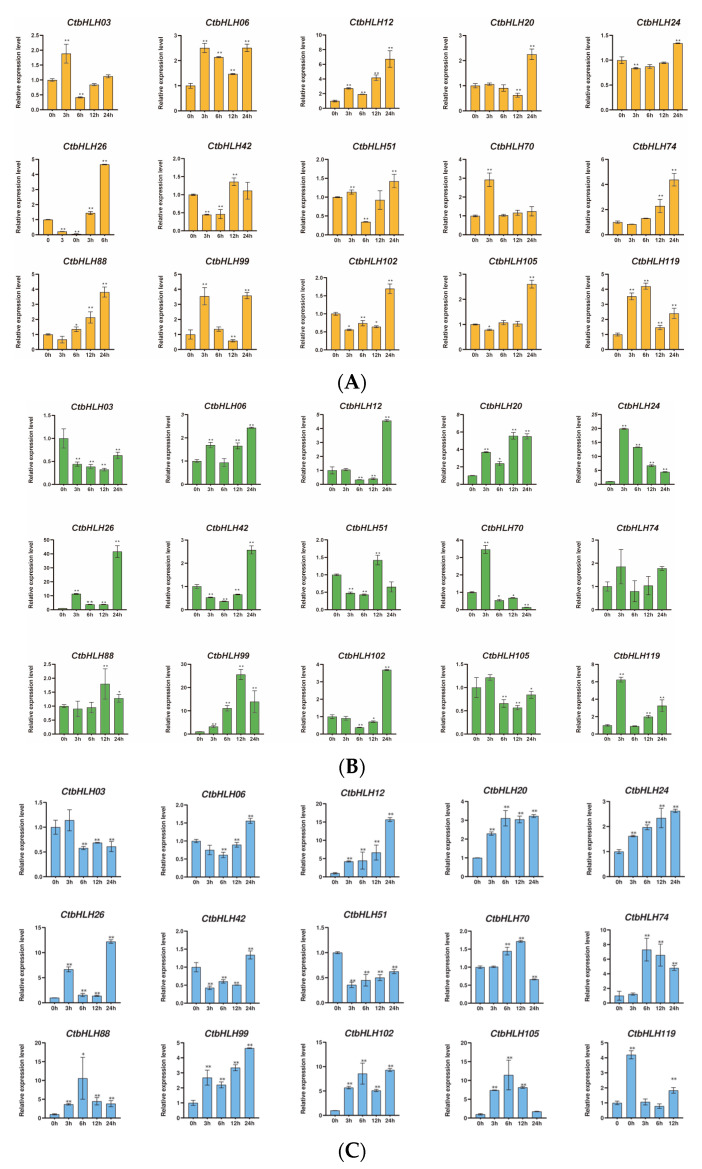
Expression analysis of *CtbHLH* genes following ABA, salt, drought, and MeJA treatments. Relative expression levels of *CtbHLH* genes following ABA (**A**), salt (**B**), drought (**C**), and MeJA (**D**) treatments. The *X*-axis represents the RNA samples from the leaves in different treatment groups at five time points, from left to right: control (0 h), ABA, salt, drought, and MeJA (3 h, 6 h, 12 h, 24 h). *Ct60S* gene was used as an internal control. The *Y*-axis represents the relative expression levels of *CtbHLH* genes using the 2^−ΔΔCt^ method. Data represents the mean ± SD of three biological replicates. Student’s *t*-test was used to determine the statistically significant levels for each treatment, where * *p* < 0.05, ** *p* < 0.01.

## Data Availability

Not applicable.

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
