# Peer review of "Genome-Wide Identification and Characterization of the bHLH Gene Family and Its Response to Abiotic Stresses in Carthamus tinctorius"

_plants, 2023, doi:10.3390/plants12213764_

Round 1
Reviewer 1 Report
Comments and Suggestions for Authors
The current research focused on the identification and expression pattern analysis of bHLH transcription factor family in C. tinctorius. First, the in silico analyses are presented. Then, the expression pattern analyses of 15 genes are reported. The objective of the manuscript is clearly defined. Unfortunately, the text is often difficult to understand, and a professional language editing service is required. In addition, there are several issues that should be addressed to improve the manuscript.
I have listed some comments and suggestions which help authors to improve their manuscript.
1- Title: I suggest “Genome-wide Identification and Characterization of the bHLH
Gene Family and Its Response to Abiotic Stresses in Carthamus tinctorius”.
2- Scientific names should be written in italics
3- L19 and along the manuscript ( including figures): 0 day should be “control”
4- I am concerned about the quality of the phylogenetic inference. Neighbor joining is not the most robust phylogenetic method, so I suggest rerunning phylogenetic analyses with a different, more robust method. I would also like to see a bootstrap analysis to provide support to clades in the tree to determine how robust the relationships present in the tree are?
5- The Ct60S gene was used as the internal reference”. Only one internal control gene was used in qPCR experiment. This gene was previously tested in similar experimental conditions and properly validated? The use of a second internal control is strongly suggested.
6- It is not clear in the text and figures if ΔΔCt or 2-ΔΔCt is statistically analyzed.
7- The visual quality of figures is low. This may just be for the images merged into the compiled file for review, but I suggest producing figures with higher DPI for better quality if that persists in the submitted figure files.
Comments on the Quality of English LanguageA professional language editing service is required
Author Response
Response to Reviewer 1 (Highlighted in green)
- Thank you for the comments. We agreed with the reviewer. The title has now been revised accordingly, L2-L4. We understand that a revised title can contribute to the overall cohesion and coherence of the manuscript.
- The text employs italics for the scientific names, with the exception of the references, which have been formatted following the referencing style guidelines of the MDPI Plant Journal.
- The manuscript has replaced occurrences of "0 days" with the term "control", e.g L386, L545 and L791.
- The neighbor-joining method has found application in numerous genomics research endeavors, notably in bHLH studies. This technique has been employed across various plant species, with safflower serving as an illustrative example, as demonstrated by Heim et al. [9], Li et al. [4], Zhao et al. [10], Song et al. [6], Sun et al. [40], and Wang et al. [43]. Furthermore, we have introduced a crucial enhancement to our analysis. We have incorporated a comprehensive set of 1000 bootstrapping analyses. These analyses serve to bolster the reliability of the clade relationships within the tree, providing a robust assessment of the relationships underpinning our findings. In addition, from the phylogenetic tree results constructed in our study, it can be seen that the classification results of all AtbHLH are the same as those of Heim et al. [9], it indicates that the phylogenetic tree constructed in this study is reliable.
- Your concern is appreciated. The selection of the Ct60S gene (60S acidic ribosomal protein) as the internal control gene was based on its established usage and thorough validation in safflower qPCR experiments, as demonstrated in prior research by Liu et al. (2016) and Tu et al. (2016). A clarification regarding the rationale for choosing this housekeeping gene has been inserted between L761-L765, accompanied by the inclusion of two additional references [101-102] at L1046-L1050."
- The reviewer's keen observation underscores the importance of addressing gene normalization in our work. Consequently, we have incorporated a comprehensive statement regarding gene expression levels and the normalization process into the text at L201-L203.
- We apologize for inadvertently submitting figures with suboptimal DPI for the review process. We have taken immediate steps to enhance the quality of all images, spanning from Figure 1 to Figure 9, ensuring that the DPI now meets the standards required for publication.

Reviewer 2 Report
Comments and Suggestions for Authors
The paper by Liang et al investigates bHLH transcription factors of Carthamus tinctorious. The paper increases knowledge about this superfamily of transcription factors that have key roles in many central processes, such as Fe uptake, biotic and abiotic stress responses, and anthocyanin biosynthesis. The majority of the study is an in silico analysis, although it also includes analysis of bHLH gene expression in plants grown under various stressful conditions.
Some changes could improve the manuscript in my opinion:
· Introduction: Despite the recognized importance of bHLH TFs in anthocyanin biosynthesis, the amount of text on this topic is too high. Furthemore, there are too many dramatic adjectives, such as enchanting, harmonious, tender, alluring, etc. In addition, it is necessary to highlight the importance of bHLH TFs in stress responses.
· Results:
-Most figures are of low quality (especially Figure 1 and Figure 8). The x axis label and letter A are missing in Figure 8. Most of the labels in Figure 8 are difficult to read. Most of Figure legends are the title of the figure, with little or none information about experimental procedures.
· Conclusions
-The conclusions focus mainly on stress responses, with only a brief mention of anthocyanin biosynthesis. This highlights the need to modify the introduction as recommended above.
Author Response
Response to Reviewer 2 (Highlighted in turquoise)
We appreciate the thorough review of our paper, and we concur with the reviewer regarding the significance of transcription factors in central processes. In response to the reviewer's suggestions, we have revised and improved the introduction section. Specifically, we have trimmed the text related to anthocyanin biosynthesis and eliminated dramatic adjectives such as 'harmonious,' 'enchanting,' 'tender,' 'alluring tapestry,' etc. Additionally, we have expanded on the importance of bHLH in stress responses, with dedicated emphasis in the introduction section (L76-L96)."
Results - We have made several important enhancements to the figures in our manuscript. Firstly, we have renumbered our figures, extending them from Figure 1 to Figure 9. Furthermore, we have taken measures to improve the overall quality of Figures 1-9. Specifically, we have added X-axis labels and description for Figure 9 to enhance its clarity. To enhance visual appeal, we have also applied new colors to Figures 9A-9D.
Additionally, we have included essential details about the experimental procedures in the legends for Figures 1-9. Consequently, we have updated the figure numbering throughout the results and discussion sections.
Conclusion - As suggested, we have made substantial improvements to the introduction section.
We appreciate the constructive feedback provided by the reviewer and have taken their comments into full consideration. We believe that these revisions align with the reviewer's recommendations and meet the necessary criteria for publication

Round 2
Reviewer 1 Report
Comments and Suggestions for Authors
Authors have made needed improvements to their manuscript. Therefore, based on the overall evaluation of the manuscript, I find it suitable for publication.
Author Response
Dear Reviewer:
Thank you for your suggestions on manuscript revision